# DVAE#: Discrete Variational Autoencoders with Relaxed Boltzmann Priors

**Arash Vahdat**[*], **Evgeny Andriyash**[*], **William G. Macready**
Quadrant.ai, D-Wave Systems Inc.
Burnaby, BC, Canada
{arash,evgeny,bill}@quadrant.ai

## Abstract

Boltzmann machines are powerful distributions that have been shown to be an effective prior over binary latent variables in variational autoencoders (VAEs). However, previous methods for training discrete VAEs have used the evidence lower bound and not the tighter importance-weighted bound. We propose two approaches for relaxing Boltzmann machines to continuous distributions that permit training with importance-weighted bounds. These relaxations are based on generalized overlapping transformations and the Gaussian integral trick. Experiments on the MNIST and OMNIGLOT datasets show that these relaxations outperform previous discrete VAEs with Boltzmann priors. An implementation which reproduces these results is available at https://github.com/QuadrantAI/dvae.

## 1 Introduction

Advances in amortized variational inference [1, 2, 3, 4] have enabled novel learning methods [4, 5, 6] and extended generative learning into complex domains such as molecule design [7, 8], music [9] and program [10] generation. These advances have been made using continuous latent variable models in spite of the computational efficiency and greater interpretability offered by discrete latent variables. Further, models such as clustering, semi-supervised learning, and variational memory addressing [11] all require discrete variables, which makes the training of discrete models an important challenge.

Prior to the deep learning era, Boltzmann machines were widely used for learning with discrete latent variables. These powerful multivariate binary distributions can represent any distribution defined on a set of binary random variables [12], and have seen application in unsupervised learning [13], supervised learning [14, 15], reinforcement learning [16], dimensionality reduction [17], and collaborative filtering [18]. Recently, Boltzmann machines have been used as priors for variational autoencoders (VAEs) in the discrete variational autoencoder (DVAE) [19] and its successor DVAE++ [20]. It has been demonstrated that these VAE models can capture discrete aspects of data. However, both these models assume a particular variational bound and tighter bounds such as the importance weighted (IW) bound [21] cannot be used for training.

We remove this constraint by introducing two continuous relaxations that convert a Boltzmann machine to a distribution over continuous random variables. These relaxations are based on overlapping transformations introduced in [20] and the Gaussian integral trick [22] (known as the Hubbard-Stratonovich transform [23] in physics). Our relaxations are made tunably sharp by using an inverse temperature parameter.

VAEs with relaxed Boltzmann priors can be trained using standard techniques developed for continuous latent variable models. In this work, we train discrete VAEs using the same IW bound on the log-likelihood that has been shown to improve importance weighted autoencoders (IWAEs) [21].

---

[*]Equal contribution

This paper makes two contributions: i) We introduce two continuous relaxations of Boltzmann machines and use these relaxations to train a discrete VAE with a Boltzmann prior using the IW bound. ii) We generalize the overlapping transformations of [20] to any pair of distributions with computable probability density function (PDF) and cumulative density function (CDF). Using these more general overlapping transformations, we propose new smoothing transformations using mixtures of Gaussian and power-function [24] distributions. Power-function overlapping transformations provide lower variance gradient estimates and improved test set log-likelihoods when the inverse temperature is large. We name our framework DVAE# because the best results are obtained when the power-function transformations are sharp.[2]

## 1.1 Related Work

Previous work on training discrete latent variable models can be grouped into five main categories:

  i) Exhaustive approaches marginalize all discrete variables [25, 26] and which are not scalable to more than a few discrete variables.

 ii) Local expectation gradients [27] and reparameterization and marginalization [28] estimators compute low-variance estimates at the cost of multiple function evaluations per gradient. These approaches can be applied to problems with a moderate number of latent variables.

iii) Relaxed computation of discrete densities [29] replaces discrete variables with continuous relaxations for gradient computation. A variation of this approach, known as the straight-through technique, sets the gradient of binary variables to the gradient of their mean [30, 31].

 iv) Continuous relaxations of discrete distributions [32] replace discrete distributions with continuous ones and optimize a consistent objective. This method cannot be applied directly to Boltzmann distributions. The DVAE [19] solves this problem by pairing each binary variable with an auxiliary continuous variable. This approach is described in Sec. 2.

  v) The REINFORCE estimator [33] (also known as the likelihood ratio [34] or score-function estimator) replaces the gradient of an expectation with the expectation of the gradient of the score function. This estimator has high variance, but many increasingly sophisticated methods provide lower variance estimators. NVIL [3] uses an input-dependent baseline, and MuProp [35] uses a first-order Taylor approximation along with an input-dependent baseline to reduce noise. VIMCO [36] trains an IWAE with binary latent variables and uses a leave-one-out scheme to define the baseline for each sample. REBAR [37] and its generalization RELAX [38] use the reparameterization of continuous distributions to define baselines.

The method proposed here is of type iv) and differs from [19, 20] in the way that binary latent variables are marginalized. The resultant relaxed distribution allows for DVAE training with a tighter bound. Moreover, our proposal encompasses a wider variety of smoothing methods and one of these empirically provides lower-variance gradient estimates.

## 2 Background

Let $\boldsymbol{x}$ represent observed random variables and $\boldsymbol{\zeta}$ continuous latent variables. We seek a generative model $p(\boldsymbol{x}, \boldsymbol{\zeta}) = p(\boldsymbol{\zeta})p(\boldsymbol{x}|\boldsymbol{\zeta})$ where $p(\boldsymbol{\zeta})$ denotes the prior distribution and $p(\boldsymbol{x}|\boldsymbol{\zeta})$ is a probabilistic decoder. In the VAE [1], training maximizes a variational lower bound on the marginal log-likelihood:

$$\log p(\boldsymbol{x}) \geq \mathbb{E}_{q(\boldsymbol{\zeta}|\boldsymbol{x})}\big[\log p(\boldsymbol{x}|\boldsymbol{\zeta})\big] - \mathrm{KL}\big(q(\boldsymbol{\zeta}|\boldsymbol{x})||p(\boldsymbol{\zeta})\big).$$

A probabilistic encoder $q(\boldsymbol{\zeta}|\boldsymbol{x})$ approximates the posterior over latent variables. For continuous $\boldsymbol{\zeta}$, the bound is maximized using the reparameterization trick. With reparameterization, expectations with respect to $q(\boldsymbol{\zeta}|\boldsymbol{x})$ are replaced by expectations against a base distribution and a differentiable function that maps samples from the base distribution to $q(\boldsymbol{\zeta}|\boldsymbol{x})$. This can always be accomplished when $q(\boldsymbol{\zeta}|\boldsymbol{x})$ has an analytic inverse cumulative distribution function (CDF) by mapping uniform samples through the inverse CDF. However, reparameterization cannot be applied to binary latent variables because the CDF is not differentiable.

The DVAE [19] resolves this issue by pairing each binary latent variable with a continuous counterpart. Denoting a binary vector of length $D$ by $\boldsymbol{z} \in \{0,1\}^D$, the Boltzmann prior is $p(\boldsymbol{z}) = e^{-E_{\boldsymbol{\theta}}(\boldsymbol{z})}/Z_{\boldsymbol{\theta}}$ where $E_{\boldsymbol{\theta}}(\boldsymbol{z}) = -\boldsymbol{a}^T \boldsymbol{z} - \frac{1}{2}\boldsymbol{z}^T \boldsymbol{W} \boldsymbol{z}$ is an energy function with parameters $\boldsymbol{\theta} \equiv \{\boldsymbol{W}, \boldsymbol{a}\}$ and partition function $Z_{\boldsymbol{\theta}}$. The joint model over discrete and continuous variables is $p(\boldsymbol{x}, \boldsymbol{z}, \boldsymbol{\zeta}) = p(\boldsymbol{z})r(\boldsymbol{\zeta}|\boldsymbol{z})p(\boldsymbol{x}|\boldsymbol{\zeta})$ where $r(\boldsymbol{\zeta}|\boldsymbol{z}) = \prod_i r(\zeta_i|z_i)$ is a smoothing transformation that maps each discrete $z_i$ to its continuous analogue $\zeta_i$.

DVAE [19] and DVAE++ [20] differ in the type of smoothing transformations $r(\zeta|z)$: [19] uses spike-and-exponential transformation (Eq. (1) left), while [20] uses two overlapping exponential distributions (Eq. (1) right). Here, $\delta(\zeta)$ is the (one-sided) Dirac delta distribution, $\zeta \in [0,1]$, and $Z_{\beta}$ is the normalization constant:

$$r(\zeta|z) = \begin{cases} \delta(\zeta) & \text{if } z = 0 \\ e^{\beta(\zeta-1)}/Z_{\beta} & \text{otherwise} \end{cases}, \qquad r(\zeta|z) = \begin{cases} e^{-\beta\zeta}/Z_{\beta} & \text{if } z = 0 \\ e^{\beta(\zeta-1)}/Z_{\beta} & \text{otherwise} \end{cases}. \qquad (1)$$

The variational bound for a factorial approximation to the posterior where $q(\boldsymbol{\zeta}|\boldsymbol{x}) = \prod_i q(\zeta_i|\boldsymbol{x})$ and $q(\boldsymbol{z}|\boldsymbol{x}) = \prod_i q(z_i|\boldsymbol{x})$ is derived in [20] as

$$\log p(\boldsymbol{x}) \geq \mathbb{E}_{q(\boldsymbol{\zeta}|\boldsymbol{x})}\left[\log p(\boldsymbol{x}|\boldsymbol{\zeta})\right] + \mathrm{H}(q(\boldsymbol{z}|\boldsymbol{x})) + \mathbb{E}_{q(\boldsymbol{\zeta}|\boldsymbol{x})}\left[\mathbb{E}_{q(\boldsymbol{z}|\boldsymbol{x},\boldsymbol{\zeta})}\log p(\boldsymbol{z})\right], \qquad (2)$$

Here $q(\zeta_i|\boldsymbol{x}) = \sum_{z_i} q(z_i|\boldsymbol{x})r(\zeta_i|z_i)$ is a mixture distribution combining $r(\zeta_i|z_i = 0)$ and $r(\zeta_i|z_i = 1)$ with weights $q(z_i|\boldsymbol{x})$. The probability of binary units conditioned on $\zeta_i$, $q(\boldsymbol{z}|\boldsymbol{x}, \boldsymbol{\zeta}) = \prod_i q(z_i|\boldsymbol{x}, \zeta_i)$, can be computed analytically. $\mathrm{H}(q(\boldsymbol{z}|\boldsymbol{x}))$ is the entropy of $q(\boldsymbol{z}|\boldsymbol{x})$. The second and third terms in Eq. (2) have analytic solutions (up to the log normalization constant) that can be differentiated easily with an automatic differentiation (AD) library. The expectation over $q(\boldsymbol{\zeta}|\boldsymbol{x})$ is approximated with reparameterized sampling.

We extend [19, 20] to tighten the bound of Eq. (2) by importance weighting [21, 39]. These tighter bounds are shown to improve VAEs. For continuous latent variables, the $K$-sample IW bound is

$$\log p(\boldsymbol{x}) \geq \mathcal{L}_K(\boldsymbol{x}) = \mathbb{E}_{\boldsymbol{\zeta}^{(k)} \sim q(\boldsymbol{\zeta}|\boldsymbol{x})}\left[\log\left(\frac{1}{K}\sum_{k=1}^{K}\frac{p(\boldsymbol{\zeta}^{(k)})p(\boldsymbol{x}|\boldsymbol{\zeta}^{(k)})}{q(\boldsymbol{\zeta}^{(k)}|\boldsymbol{x})}\right)\right]. \qquad (3)$$

The tightness of the IW bound improves as $K$ increases [21].

## 3 Model

We introduce two relaxations of Boltzmann machines to define the continuous prior distribution $p(\boldsymbol{\zeta})$ in the IW bound of Eq. (3). These relaxations rely on either overlapping transformations (Sec. 3.1) or the Gaussian integral trick (Sec. 3.2). Sec. 3.3 then generalizes the class of overlapping transformations that can be used in the approximate posterior $q(\boldsymbol{\zeta}|\boldsymbol{x})$.

### 3.1 Overlapping Relaxations

We obtain a continuous relaxation of $p(\boldsymbol{z})$ through the marginal $p(\boldsymbol{\zeta}) = \sum_z p(\boldsymbol{z})r(\boldsymbol{\zeta}|\boldsymbol{z})$ where $r(\boldsymbol{\zeta}|\boldsymbol{z})$ is an overlapping smoothing transformation [20] that operates on each component of $\boldsymbol{z}$ and $\boldsymbol{\zeta}$ independently; i.e., $r(\boldsymbol{\zeta}|\boldsymbol{z}) = \prod_i r(\zeta_i|z_i)$. Overlapping transformations such as mixture of exponential in Eq. (1) may be used for $r(\boldsymbol{\zeta}|\boldsymbol{z})$. These transformations are equipped with an inverse temperature hyperparameter $\beta$ to control the sharpness of the smoothing transformation. As $\beta \to \infty$, $r(\boldsymbol{\zeta}|\boldsymbol{z})$ approaches $\delta(\boldsymbol{\zeta} - \boldsymbol{z})$ and $p(\boldsymbol{\zeta}) = \sum_z p(\boldsymbol{z})\delta(\boldsymbol{\zeta} - \boldsymbol{z})$ becomes a mixture of $2^D$ delta function distributions centered on the vertices of the hypercube in $\mathbb{R}^D$. At finite $\beta$, $p(\boldsymbol{\zeta})$ provides a continuous relaxation of the Boltzmann machine.

To train an IWAE using Eq. (3) with $p(\boldsymbol{\zeta})$ as a prior, we must compute $\log p(\boldsymbol{\zeta})$ and its gradient with respect to the parameters of the Boltzmann distribution and the approximate posterior. This computation involves marginalization over $\boldsymbol{z}$, which is generally intractable. However, we show that this marginalization can be approximated accurately using a mean-field model.

#### 3.1.1 Computing $\log p(\boldsymbol{\zeta})$ and its Gradient for Overlapping Relaxations

Since overlapping transformations are factorial, the log marginal distribution of $\boldsymbol{\zeta}$ is

$$\log p(\boldsymbol{\zeta}) = \log\left(\sum_{\boldsymbol{z}} p(\boldsymbol{z})r(\boldsymbol{\zeta}|\boldsymbol{z})\right) = \log\left(\sum_{\boldsymbol{z}} e^{-E_{\boldsymbol{\theta}}(\boldsymbol{z}) + \boldsymbol{b}^{\beta}(\boldsymbol{\zeta})^T \boldsymbol{z} + \boldsymbol{c}^{\beta}(\boldsymbol{\zeta})}\right) - \log Z_{\boldsymbol{\theta}}, \qquad (4)$$

where $b_i^\beta(\boldsymbol{\zeta}) = \log r(\zeta_i|z_i = 1) - \log r(\zeta_i|z_i = 0)$ and $c_i^\beta(\boldsymbol{\zeta}) = \log r(\zeta_i|z_i = 0)$. For the mixture of exponential smoothing $b_i^\beta(\boldsymbol{\zeta}) = \beta(2\zeta_i - 1)$ and $c_i^\beta(\boldsymbol{\zeta}) = -\beta\zeta_i - \log Z_\beta$.

The first term in Eq. (4) is the log partition function of the Boltzmann machine $\hat{p}(\boldsymbol{z})$ with augmented energy function $\hat{E}_{\boldsymbol{\theta},\boldsymbol{\zeta}}^\beta(\boldsymbol{z}) := E_{\boldsymbol{\theta}}(\boldsymbol{z}) - \boldsymbol{b}^\beta(\boldsymbol{\zeta})^T\boldsymbol{z} - \boldsymbol{c}^\beta(\boldsymbol{\zeta})$. Estimating the log partition function accurately can be expensive, particularly because it has to be done for each $\boldsymbol{\zeta}$. However, we note that each $\zeta_i$ comes from a bimodal distribution centered at zero and one, and that the bias $\boldsymbol{b}^\beta(\boldsymbol{\zeta})$ is usually large for most components $i$ (particularly for large $\beta$). In this case, mean field is likely to provide a good approximation of $\hat{p}(\boldsymbol{z})$, a fact we demonstrate empirically in Sec. 4.

To compute $\log p(\boldsymbol{\zeta})$ and its gradient, we first fit a mean-field distribution $m(\boldsymbol{z}) = \prod_i m_i(z_i)$ by minimizing $\mathrm{KL}(m(\boldsymbol{z})||\hat{p}(\boldsymbol{z}))$ [40]. The gradient of $\log p(\boldsymbol{\zeta})$ with respect to $\beta$, $\boldsymbol{\theta}$ or $\boldsymbol{\zeta}$ is:

$$
\begin{aligned}
\nabla \log p(\boldsymbol{\zeta}) &= -\mathbb{E}_{\boldsymbol{z}\sim\hat{p}(\boldsymbol{z})}\big[\nabla \hat{E}_{\boldsymbol{\theta},\boldsymbol{\zeta}}^\beta(\boldsymbol{z})\big] + \mathbb{E}_{\boldsymbol{z}\sim p(\boldsymbol{z})}\big[\nabla E_{\boldsymbol{\theta}}(\boldsymbol{z})\big] \\
&\approx -\mathbb{E}_{\boldsymbol{z}\sim m(\boldsymbol{z})}\big[\nabla \hat{E}_{\boldsymbol{\theta},\boldsymbol{\zeta}}^\beta(\boldsymbol{z})\big] + \mathbb{E}_{\boldsymbol{z}\sim p(\boldsymbol{z})}\big[\nabla E_{\boldsymbol{\theta}}(\boldsymbol{z})\big] \\
&= -\nabla \hat{E}_{\boldsymbol{\theta},\boldsymbol{\zeta}}^\beta(\boldsymbol{m}) + \mathbb{E}_{\boldsymbol{z}\sim p(\boldsymbol{z})}\big[\nabla E_{\boldsymbol{\theta}}(\boldsymbol{z})\big],
\end{aligned}
\tag{5}
$$

where $\boldsymbol{m}^T = [m_1(z_1 = 1) \quad \cdots \quad m_D(z_D = 1)]$ is the mean-field solution and where the gradient does not act on $\boldsymbol{m}$. The first term in Eq. (5) is the result of computing the average energy under a factorial distribution.[3] The second expectation corresponds to the negative phase in training Boltzmann machines and is approximated by Monte Carlo sampling from $p(\boldsymbol{z})$.

To compute the importance weights for the IW bound of Eq. (3) we must compute the value of $\log p(\boldsymbol{\zeta})$ up to the normalization; i.e. the first term in Eq. (4). Assuming that $KL\big(m(\boldsymbol{z})||\hat{p}(\boldsymbol{z})\big) \approx 0$ and using

$$
\mathrm{KL}(m(\boldsymbol{z})||\hat{p}(\boldsymbol{z})) = \hat{E}_{\boldsymbol{\theta},\boldsymbol{\zeta}}^\beta(\boldsymbol{m}) + \log\Big(\sum_z e^{-\hat{E}_{\boldsymbol{\theta},\boldsymbol{\zeta}}^\beta(\boldsymbol{z})}\Big) - \mathrm{H}(m(\boldsymbol{z})),
\tag{6}
$$

the first term of Eq. (4) is approximated as $\mathrm{H}\big(m(\boldsymbol{z})\big) - \hat{E}_{\boldsymbol{\theta},\boldsymbol{\zeta}}^\beta(\boldsymbol{m})$.

## 3.2 The Gaussian Integral Trick

The computational complexity of $\log p(\boldsymbol{\zeta})$ arises from the pairwise interactions $\boldsymbol{z}^T\boldsymbol{W}\boldsymbol{z}$ present in $E_{\boldsymbol{\theta}}(\boldsymbol{z})$. Instead of applying mean field, we remove these interactions using the Gaussian integral trick [41]. This is achieved by defining Gaussian smoothing:

$$
r(\boldsymbol{\zeta}|\boldsymbol{z}) = \mathcal{N}(\boldsymbol{\zeta}|\boldsymbol{A}(\boldsymbol{W} + \beta\boldsymbol{I})\boldsymbol{z}, \boldsymbol{A}(\boldsymbol{W} + \beta\boldsymbol{I})\boldsymbol{A}^T)
$$

for an invertible matrix $\boldsymbol{A}$ and a diagonal matrix $\beta\boldsymbol{I}$ with $\beta > 0$. Here, $\beta$ must be large enough so that $\boldsymbol{W} + \beta\boldsymbol{I}$ is positive definite. Common choices for $\boldsymbol{A}$ include $\boldsymbol{A} = \boldsymbol{I}$ or $\boldsymbol{A} = \boldsymbol{\Lambda}^{-\frac{1}{2}}\boldsymbol{V}^T$ where $\boldsymbol{V}\boldsymbol{\Lambda}\boldsymbol{V}^T$ is the eigendecomposition of $\boldsymbol{W} + \beta\boldsymbol{I}$ [41]. However, neither of these choices places the modes of $p(\boldsymbol{\zeta})$ on the vertices of the hypercube in $\mathbb{R}^D$. Instead, we take $\boldsymbol{A} = (\boldsymbol{W} + \beta\boldsymbol{I})^{-1}$ giving the smoothing transformation $r(\boldsymbol{\zeta}|\boldsymbol{z}) = \mathcal{N}(\boldsymbol{\zeta}|\boldsymbol{z}, (\boldsymbol{W} + \beta\boldsymbol{I})^{-1})$. The joint density is then

$$
p(\boldsymbol{z}, \boldsymbol{\zeta}) \propto e^{-\frac{1}{2}\boldsymbol{\zeta}^T(\boldsymbol{W}+\beta\boldsymbol{I})\boldsymbol{\zeta} + \boldsymbol{z}^T(\boldsymbol{W}+\beta\boldsymbol{I})\boldsymbol{\zeta} + (\boldsymbol{a}-\frac{1}{2}\beta\boldsymbol{1})^T\boldsymbol{z}},
$$

where $\boldsymbol{1}$ is the $D$-vector of all ones. Since $p(\boldsymbol{z}, \boldsymbol{\zeta})$ no longer contains pairwise interactions $\boldsymbol{z}$ can be marginalized out giving

$$
p(\boldsymbol{\zeta}) = Z_{\boldsymbol{\theta}}^{-1}\left|\frac{1}{2\pi}(\boldsymbol{W} + \beta\boldsymbol{I})\right|^{\frac{1}{2}} e^{-\frac{1}{2}\boldsymbol{\zeta}^T(\boldsymbol{W}+\beta\boldsymbol{I})\boldsymbol{\zeta}} \prod_i \Big[1 + e^{a_i + c_i - \frac{\beta}{2}}\Big],
\tag{7}
$$

where $c_i$ is the $i^{\text{th}}$ element of $(\boldsymbol{W} + \beta\boldsymbol{I})\boldsymbol{\zeta}$.

The marginal $p(\boldsymbol{\zeta})$ in Eq. (7) is a mixture of $2^D$ Gaussian distributions centered on the vertices of the hypercube in $\mathbb{R}^D$ with mixing weights given by $p(\boldsymbol{z})$. Each mixture component has covariance $\boldsymbol{\Sigma} = (\boldsymbol{W} + \beta\boldsymbol{I})^{-1}$ and, as $\beta$ gets large, the precision matrix becomes diagonally dominant. As

$\beta \to \infty$, each mixture component becomes a delta function and $p(\boldsymbol{\zeta})$ approaches $\sum_z p(\boldsymbol{z})\delta(\boldsymbol{\zeta} - \boldsymbol{z})$. This Gaussian smoothing allows for simple evaluation of $\log p(\boldsymbol{\zeta})$ (up to $Z_{\boldsymbol{\theta}}$), but we note that each mixture component has a nondiagonal covariance matrix, which should be accommodated when designing the approximate posterior $q(\boldsymbol{\zeta}|\boldsymbol{x})$.

The hyperparameter $\beta$ must be larger than the absolute value of the most negative eigenvalue of $\boldsymbol{W}$ to ensure that $\boldsymbol{W} + \beta\boldsymbol{I}$ is positive definite. Setting $\beta$ to even larger values has the benefit of making the Gaussian mixture components more isotropic, but this comes at the cost of requiring a sharper approximate posterior with potentially noisier gradient estimates.

### 3.3   Generalizing Overlapping Transformations

The previous sections developed two $r(\boldsymbol{\zeta}|\boldsymbol{z})$ relaxations for Boltzmann priors. Depending on this choice, compatible $q(\boldsymbol{\zeta}|\boldsymbol{x})$ parameterizations must be used. For example, if Gaussian smoothing is used, then a mixture of Gaussian smoothers should be used in the approximate posterior. Unfortunately, the overlapping transformations introduced in DVAE++ [20] are limited to mixtures of exponential or logistic distributions where the inverse CDF can be computed analytically. Here, we provide a general approach for reparameterizing overlapping transformations that does not require analytic inverse CDFs. Our approach is a special case of the reparameterization method for multivariate mixture distributions proposed in [42].

Assume $q(\zeta|\boldsymbol{x}) = (1 - q)r(\zeta|z = 0) + qr(\zeta|z = 1)$ is the mixture distribution resulting from an overlapping transformation defined for one-dimensional $z$ and $\zeta$ where $q \equiv q(z = 1|\boldsymbol{x})$. Ancestral sampling from $q(\zeta|\boldsymbol{x})$ is accomplished by first sampling from the binary distribution $q(z|\boldsymbol{x})$ and then sampling $\zeta$ from $r(\zeta|z)$. This process generates samples but is not differentiable with respect to $q$.

To compute the gradient (with respect to $q$) of samples from $q(\zeta|\boldsymbol{x})$, we apply the implicit function theorem. The inverse CDF of $q(\zeta|\boldsymbol{x})$ at $\rho$ is obtained by solving:

$$\text{CDF}(\zeta) = (1 - q)R(\zeta|z = 0) + qR(\zeta|z = 1) = \rho, \tag{8}$$

where $\rho \in [0, 1]$ and $R(\zeta|z)$ is the CDF for $r(\zeta|z)$. Assuming that $\zeta$ is a function of $q$ but $\rho$ is not, we take the gradient from both sides of Eq. (8) with respect to $q$ giving

$$\frac{\partial \zeta}{\partial q} = \frac{R(\zeta|z = 0) - R(\zeta|z = 1)}{(1 - q)r(\zeta|z = 0) + qr(\zeta|z = 1)}, \tag{9}$$

which can be easily computed for a sampled $\zeta$ if the PDF and CDF of $r(\zeta|z)$ are known. This generalization allows us to compute gradients of samples generated from a wide range of overlapping transformations. Further, the gradient of $\zeta$ with respect to the parameters of $r(\zeta|z)$ (e.g. $\beta$) is computed similarly as

$$\frac{\partial \zeta}{\partial \beta} = -\frac{(1 - q)\,\partial_\beta R(\zeta|z = 0) + q\,\partial_\beta R(\zeta|z = 1)}{(1 - q)r(\zeta|z = 0) + qr(\zeta|z = 1)}.$$

With this method, we can apply overlapping transformations beyond the mixture of exponentials considered in [20]. The inverse CDF of exponential mixtures is shown in Fig. 1(a) for several $\beta$. As $\beta$ increases, the relaxation approaches the original binary variables, but this added fidelity comes at the cost of noisy gradients. Other overlapping transformations offer alternative tradeoffs:

**Uniform+Exp Transformation:** We ensure that the gradient remains finite as $\beta \to \infty$ by mixing the exponential with a uniform distribution. This is achieved by defining $r'(\zeta|z) = (1 - \epsilon)r(\zeta|z) + \epsilon$ where $r(\zeta|z)$ is the exponential smoothing and $\zeta \in [0, 1]$. The inverse CDF resulting from this smoothing is shown in Fig. 1(b).

**Power-Function Transformation:** Instead of adding a uniform distribution we substitute the exponential distribution for one with heavier tails. One choice is the power-function distribution [24]:

$$r(\zeta|z) = \begin{cases} \frac{1}{\beta}\zeta^{(\frac{1}{\beta}-1)} & \text{if } z = 0 \\ \frac{1}{\beta}(1 - \zeta)^{(\frac{1}{\beta}-1)} & \text{otherwise} \end{cases} \quad \text{for } \zeta \in [0, 1] \text{ and } \beta > 1. \tag{10}$$

The conditionals in Eq. (10) correspond to the Beta distributions $B(1/\beta, 1)$ and $B(1, 1/\beta)$ respectively. The inverse CDF resulted from this smoothing is visualized in Fig. 1(c).

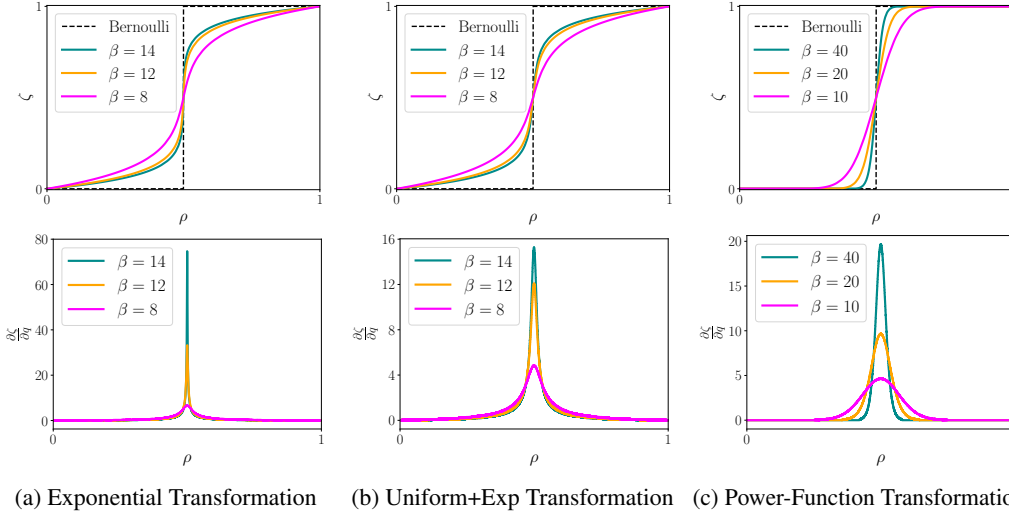

(a) Exponential Transformation     (b) Uniform+Exp Transformation   (c) Power-Function Transformation

Figure 1: In the first row, we visualize the inverse CDF of the mixture $q(\zeta) = \sum_z q(z)r(\zeta|z)$ for $q = q(z = 1) = 0.5$ as a function of the random noise $\rho \in [0, 1]$. In the second row, the gradient of the inverse CDF with respect to $q$ is visualized. Each column corresponds to a different smoothing transformation. As the transition region sharpens with increasing $\beta$, a sampling based estimate of the gradient becomes noisier; i.e., the variance of $\partial\zeta/\partial q$ increases. The uniform+exp exponential has a very similar inverse CDF (first row) to the exponential but has potentially lower variance (bottom row). In comparison, the power-function smoothing with $\beta = 40$ provides a good relaxation of the discrete variables while its gradient noise is still moderate. See the supplementary material for a comparison of the gradient noise.

**Gaussian Transformations:** The transformations introduced above have support $\zeta \in [0, 1]$. We also explore Gaussian smoothing $r(\zeta|z) = \mathcal{N}(\zeta|z, \frac{1}{\beta})$ with support $\zeta \in \mathbb{R}$.

None of these transformations have an analytic inverse CDF for $q(\zeta|\boldsymbol{x})$ so we use Eq. (9) to calculate gradients.

## 4 Experiments

In this section we compare the various relaxations for training DVAEs with Boltzmann priors on statically binarized MNIST [43] and OMNIGLOT [44] datasets. For all experiments we use a generative model of the form $p(\boldsymbol{x}, \boldsymbol{\zeta}) = p(\boldsymbol{\zeta})p(\boldsymbol{x}|\boldsymbol{\zeta})$ where $p(\boldsymbol{\zeta})$ is a continuous relaxation obtained from either the overlapping relaxation of Eq. (4) or the Gaussian integral trick of Eq. (7). The underlying Boltzmann distribution is a restricted Boltzmann machine (RBM) with bipartite connectivity which allows for parallel Gibbs updates. We use a hierarchical autoregressively-structured $q(\boldsymbol{\zeta}|\boldsymbol{x}) = \prod_{g=1}^{G} q(\boldsymbol{\zeta}_g|\boldsymbol{x}, \boldsymbol{\zeta}_{<g})$ to approximate the posterior distribution over $\boldsymbol{\zeta}$. This structure divides the components of $\boldsymbol{\zeta}$ into $G$ equally-sized groups and defines each conditional using a factorial distribution conditioned on $\boldsymbol{x}$ and all $\boldsymbol{\zeta}$ from previous groups.

The smoothing transformation used in $q(\boldsymbol{\zeta}|\boldsymbol{x})$ depends on the type of relaxation used in $p(\boldsymbol{\zeta})$. For overlapping relaxations, we compare exponential, uniform+exp, Gaussian, and power-function. With the Gaussian integral trick, we use shifted Gaussian smoothing as described below. The decoder $p(\boldsymbol{x}|\boldsymbol{\zeta})$ and conditionals $q(\boldsymbol{\zeta}_g|\boldsymbol{x}, \boldsymbol{\zeta}_{<g})$ are modeled with neural networks. Following [20], we consider both linear (——) and nonlinear ($\sim$) versions of these networks. The linear models use a single linear layer to predict the parameters of the distributions $p(\boldsymbol{x}|\boldsymbol{\zeta})$ and $q(\boldsymbol{\zeta}_g|\boldsymbol{x}, \boldsymbol{\zeta}_{<g})$ given their input. The nonlinear models use two deterministic hidden layers with 200 units, tanh activation and batch-normalization. We use the same initialization scheme, batch-size, optimizer, number of training iterations, schedule of learning rate, weight decay and KL warm-up for training that was used in [20] (See Sec. 7.2 in [20]). For the mean-field optimization, we use 5 iterations. To evaluate the trained models, we estimate the log-likelihood on the discrete graphical model using the importance-weighted

bound with 4000 samples [21]. At evaluation $p(\boldsymbol{\zeta})$ is replaced with the Boltzmann distribution $p(\boldsymbol{z})$, and $q(\boldsymbol{\zeta}|\boldsymbol{x})$ with $q(\boldsymbol{z}|\boldsymbol{x})$ (corresponding to $\beta = \infty$).

For DVAE, we use the original spike-and-exp smoothing. For DVAE++, in addition to exponential smoothing, we use a mixture of power-functions. The DVAE# models are trained using the IW bound in Eq. (3) with $K = 1, 5, 25$ samples. To fairly compare DVAE# with DVAE and DVAE++ (which can only be trained with the variational bound), we use the same number of samples $K \geq 1$ when estimating the variational bound during DVAE and DVAE++ training.

The smoothing parameter $\beta$ is fixed throughout training (i.e. $\beta$ is not annealed). However, since $\beta$ acts differently for each smoothing function $r$, its value is selected by cross validation per smoothing and structure. We select from $\beta \in \{4, 5, 6, 8\}$ for spike-and-exp, $\beta \in \{8, 10, 12, 16\}$ for exponential, $\beta \in \{16, 20, 30, 40\}$ with $\epsilon = 0.05$ for uniform+exp, $\beta \in \{15, 20, 30, 40\}$ for power-function, and $\beta \in \{20, 25, 30, 40\}$ for Gaussian smoothing. For models other than the Gaussian integral trick, $\beta$ is set to the same value in $q(\boldsymbol{\zeta}|\boldsymbol{x})$ and $p(\boldsymbol{\zeta})$. For the Gaussian integral case, $\beta$ in the encoder is trained as discussed next, but is selected in the prior from $\beta \in \{20, 25, 30, 40\}$.

With the Gaussian integral trick, each mixture component in the prior contains off-diagonal correlations and the approximation of the posterior over $\boldsymbol{\zeta}$ should capture this. We recall that a multivariate Gaussian $\mathcal{N}(\boldsymbol{\zeta}|\boldsymbol{\mu}, \boldsymbol{\Sigma})$ can always be represented as a product of Gaussian conditionals $\prod_i \mathcal{N}\big(\zeta_i|\mu_i + \Delta\mu_i(\boldsymbol{\zeta}_{<i}), \sigma_i\big)$ where $\Delta\mu_i(\boldsymbol{\zeta}_{<i})$ is linear in $\boldsymbol{\zeta}_{<i}$. Motivated by this observation, we provide flexibility in the approximate posterior $q(\boldsymbol{\zeta}|\boldsymbol{x})$ by using shifted Gaussian smoothing where $r(\zeta_i|z_i) = \mathcal{N}(\zeta_i|z_i + \Delta\mu_i(\boldsymbol{\zeta}_{<i}), 1/\beta_i)$, and $\Delta\mu_i(\boldsymbol{\zeta}_{<i})$ is an additional parameter that shifts the distribution. As the approximate posterior in our model is hierarchical, we generate $\Delta\mu_i(\boldsymbol{\zeta}_{<g})$ for the $i^{th}$ element in $g^{th}$ group as the output of the same neural network that generates the parameters of $q(\boldsymbol{\zeta}_g|\boldsymbol{x}, \boldsymbol{\zeta}_{<g})$. The parameter $\beta_i$ for each component of $\boldsymbol{\zeta}_g$ is a trainable parameter shared for all $\boldsymbol{x}$.

Training also requires sampling from the discrete RBM to compute the $\boldsymbol{\theta}$-gradient of $\log Z_{\boldsymbol{\theta}}$. We have used both population annealing [45] with 40 sweeps across variables per parameter update and persistent contrastive divergence [46] for sampling. Population annealing usually results in a better generative model (see the supplementary material for a comparison). We use QuPA[4], a GPU implementation of population annealing. To obtain test set log-likelihoods we require $\log Z_{\boldsymbol{\theta}}$, which we estimate with annealed importance sampling [47, 48]. We use 10,000 temperatures and 1,000 samples to ensure that the standard deviation of the $\log Z_{\boldsymbol{\theta}}$ estimate is small ($\sim 0.01$).

We compare the performance of DVAE# against DVAE and DVAE++ in Table 1. We consider four neural net structures when examining the various smoothing models. Each structure is denoted "G —/$\sim$" where G represent the number of groups in the approximate posterior and —/$\sim$ indicates linear/nonlinear conditionals. The RBM prior for the structures "1 —/$\sim$" is $100 \times 100$ (i.e. $D = 200$) and for structures "2/4 $\sim$" the RBM is $200 \times 200$ (i.e. $D = 400$).

We make several observations based on Table 1: i) Most baselines improve as $K$ increases. The improvements are generally larger for DVAE# as they optimize the IW bound. ii) Power-function smoothing improves the performance of DVAE++ over the original exponential smoothing. iii) DVAE# and DVAE++ both with power-function smoothing for $K = 1$ optimizes a similar variational bound with same smoothing transformation. The main difference here is that DVAE# uses the marginal $p(\boldsymbol{\zeta})$ in the prior whereas DVAE++ has the joint $p(\boldsymbol{z}, \boldsymbol{\zeta}) = p(\boldsymbol{z})r(\boldsymbol{z}|\boldsymbol{\zeta})$. For this case, it can be seen that DVAE# usually outperforms DVAE++ . iv) Among the DVAE# variants, the Gaussian integral trick and Gaussian overlapping relaxation result in similar performance, and both are usually inferior to the other DVAE# relaxations. v) In DVAE#, the uniform+exp smoothing performs better than exponential smoothing alone. vi) DVAE# with the power-function smoothing results in the best generative models, and in most cases outperforms both DVAE and DVAE++.

Given the superior performance of the models obtained using the mean-field approximation of Sec. 3.1.1 to $\hat{p}(\boldsymbol{\zeta})$, we investigate the accuracy of this approximation. In Fig. 2(a), we show that the mean-field model converges quickly by plotting the KL divergence of Eq. (6) with the number of mean-field iterations for a single $\boldsymbol{\zeta}$. To assess the quality of the mean-field approximation, in Fig. 2(b) we compute the KL divergence for randomly selected $\boldsymbol{\zeta}$s during training at different iterations for exponential and power-function smoothings with different $\beta$s. As it can be seen, throughout the

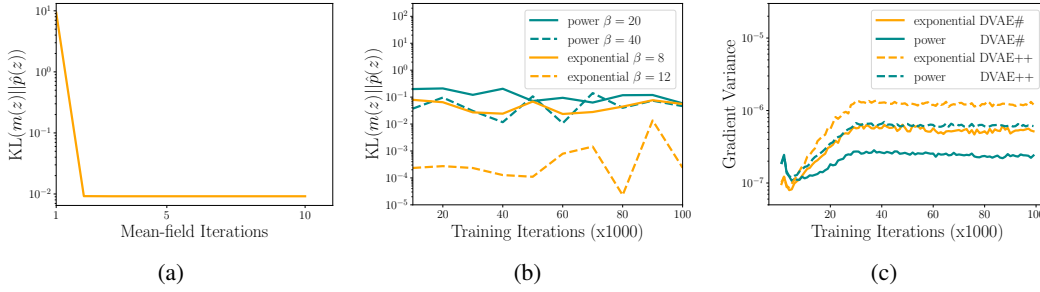

Figure 2: (a) The KL divergence between the mean-field model and the augmented Boltzmann machine $\hat{p}(\boldsymbol{z})$ as a function of the number of optimization iterations of the mean-field. The mean-field model converges to KL = 0.007 in three iterations. (b) The KL value is computed for randomly selected $\zeta$s during training at different iterations for exponential and power-function smoothings with different $\beta$. (c) The variance of the gradient of the objective function with respect to the logit of $q$ is visualized for exponential and power-function smoothing transformations. Power-function smoothing tends to have lower variance than exponential smoothing. The artifact seen early in training is due to the warm-up of KL. Models in (b) and (c) are trained for 100K iterations with batch size of 1,000.

Table 1: The performance of DVAE# is compared against DVAE and DVAE++ on MNIST and OMNIGLOT. Mean±standard deviation of the negative log-likelihood for five runs are reported.

| | Struct. | K | DVAE Spike-Exp | DVAE++ Exp | Power | DVAE# Gauss. Int | Gaussian | Exp | Un+Exp | Power |
|---|---|---|---|---|---|---|---|---|---|---|
| **MNIST** | 1 — | 1 | **89.00±0.09** | 90.43±0.06 | **89.12±0.05** | 92.14±0.12 | 91.33±0.13 | 90.55±0.11 | 89.57±0.08 | 89.35±0.06 |
| | | 5 | 89.15±0.12 | 90.13±0.03 | 89.09±0.05 | 91.32±0.09 | 90.15±0.04 | 89.62±0.08 | 88.56±0.04 | **88.25±0.03** |
| | | 25 | 89.20±0.13 | 89.92±0.07 | 89.04±0.07 | 91.18±0.21 | 89.55±0.10 | 89.27±0.09 | 88.02±0.04 | **87.67±0.07** |
| | 1 ∼ | 1 | 85.48±0.06 | 85.13±0.06 | 85.05±0.02 | 86.23±0.05 | 86.24±0.05 | 85.37±0.05 | 85.19±0.05 | **84.93±0.02** |
| | | 5 | 85.29±0.03 | 85.13±0.09 | 85.29±0.10 | 84.99±0.03 | 84.91±0.07 | 84.83±0.03 | 84.47±0.02 | **84.21±0.02** |
| | | 25 | 85.92±0.10 | 86.14±0.18 | 85.59±0.10 | 84.36±0.04 | 84.30±0.04 | 84.69±0.08 | 84.22±0.01 | **83.93±0.06** |
| | 2 ∼ | 1 | 83.97±0.04 | 84.15±0.07 | 83.62±0.04 | 84.30±0.05 | 84.35±0.04 | 83.96±0.06 | 83.54±0.06 | **83.37±0.02** |
| | | 5 | 83.74±0.03 | 84.85±0.13 | 83.57±0.07 | 83.68±0.02 | 83.61±0.04 | 83.70±0.04 | 83.33±0.04 | **82.99±0.04** |
| | | 25 | 84.19±0.21 | 85.49±0.12 | 83.58±0.15 | 83.39±0.04 | 83.26±0.04 | 83.76±0.04 | 83.30±0.04 | **82.85±0.03** |
| | 4 ∼ | 1 | 84.38±0.03 | 84.63±0.11 | 83.44±0.05 | 84.59±0.06 | 84.81±0.19 | 84.06±0.06 | 83.52±0.06 | **83.18±0.05** |
| | | 5 | 83.93±0.07 | 85.41±0.09 | 83.17±0.09 | 83.89±0.09 | 84.20±0.15 | 84.15±0.05 | 83.41±0.04 | **82.95±0.07** |
| | | 25 | 84.12±0.05 | 85.42±0.07 | 83.20±0.08 | 83.52±0.06 | 83.80±0.04 | 84.22±0.13 | 83.39±0.04 | **82.82±0.02** |
| **OMNIGLOT** | 1 — | 1 | **105.11±0.11** | 106.71±0.08 | 105.45±0.08 | 110.81±0.32 | 106.81±0.07 | 107.21±0.14 | 105.89±0.06 | 105.47±0.09 |
| | | 5 | **104.68±0.21** | 106.83±0.09 | 105.34±0.05 | 112.26±0.70 | 106.16±0.11 | 106.86±0.10 | 104.94±0.05 | **104.42±0.09** |
| | | 25 | 104.38±0.15 | 106.85±0.07 | 105.38±0.14 | 111.92±0.30 | 105.75±0.10 | 106.88±0.09 | 104.49±0.07 | **103.98±0.05** |
| | 1 ∼ | 1 | 102.95±0.07 | 101.84±0.08 | 101.88±0.06 | 103.50±0.06 | 102.74±0.08 | 102.23±0.08 | 101.86±0.06 | **101.70±0.01** |
| | | 5 | 102.45±0.08 | 102.13±0.11 | 101.67±0.07 | 102.15±0.04 | 102.00±0.09 | 101.59±0.06 | 101.22±0.05 | **101.00±0.02** |
| | | 25 | 102.74±0.05 | 102.66±0.09 | 101.80±0.15 | 101.42±0.04 | 101.60±0.09 | 101.48±0.04 | 100.93±0.07 | **100.60±0.05** |
| | 2 ∼ | 1 | 103.10±0.31 | 101.34±0.04 | 100.42±0.03 | 102.07±0.16 | 102.84±0.23 | 100.38±0.09 | **99.84±0.06** | 99.75±0.05 |
| | | 5 | 100.88±0.13 | 100.55±0.09 | 99.51±0.05 | 100.85±0.02 | 101.43±0.11 | 99.93±0.07 | 99.57±0.06 | **99.24±0.05** |
| | | 25 | 100.55±0.08 | 100.31±0.15 | 99.49±0.07 | 100.20±0.02 | 100.45±0.08 | 100.10±0.28 | 99.59±0.16 | **98.93±0.05** |
| | 4 ∼ | 1 | 104.63±0.47 | 101.58±0.22 | 100.42±0.08 | 102.91±0.25 | 103.43±0.10 | 100.85±0.12 | 99.92±0.11 | **99.65±0.09** |
| | | 5 | 101.77±0.20 | 101.01±0.09 | 99.52±0.09 | 101.79±0.25 | 101.82±0.13 | 100.32±0.19 | 99.61±0.07 | **99.13±0.10** |
| | | 25 | 100.89±0.13 | 100.37±0.09 | 99.43±0.14 | 100.73±0.08 | 100.97±0.21 | 99.92±0.30 | 99.36±0.09 | **98.88±0.09** |

training the KL value is typically $< 0.2$. For larger $\beta$s, the KL value is smaller due to the stronger bias that $\boldsymbol{b}^{\beta}(\boldsymbol{\zeta})$ imposes on $\boldsymbol{z}$.

Lastly, we demonstrate that the lower variance of power-function smoothing may contribute to its success. As noted in Fig. 1, power-function smoothing potentially has moderate gradient noise while still providing a good approximation of binary variables at large $\beta$. We validate this hypothesis in Fig. 2(c) by measuring the variance of the derivative of the variational bound (with $K = 1$) with respect to the logit of $q$ during training of a 2-layer nonlinear model on MNIST. When comparing the exponential ($\beta = 10$) to power-function smoothing ($\beta = 30$) at the $\beta$ that performs best for each smoothing method, we find that power-function smoothing has significantly lower variance.

# 5 Conclusions

We have introduced two approaches for relaxing Boltzmann machines to continuous distributions, and shown that the resulting distributions can be trained as priors in DVAEs using an importance-weighted bound. We have proposed a generalization of overlapping transformations that removes the need for computing the inverse CDF analytically. Using this generalization, the mixture of power-function smoothing provides a good approximation of binary variables while the gradient noise remains moderate. In the case of sharp power smoothing, our model outperforms previous discrete VAEs.

## Footnotes

[2]And not because our model is proposed after DVAE and DVAE++.

[3]The augmented energy $\hat{E}_{\boldsymbol{\theta},\boldsymbol{\zeta}}^\beta(\boldsymbol{z})$ is a multi-linear function of $\{z_i\}$ and under the mean-field assumption each $z_i$ is replaced by its average value $m(z_i = 1)$.

[4]This library is publicly available at https://try.quadrant.ai/qupa

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

# A    Population Annealing vs. Persistence Contrastive Divergence

In this section, we compare population annealing (PA) to persistence contrastive divergence (PCD) for sampling in the negative phase. In Table 2, we train DVAE# with the power-function smoothing on the binarized MNIST dataset using PA and PCD. As shown, PA results in a comparable generative model when there is one group of latent variables and better models in other cases.

Table 2: The performance of DVAE# with power-function smoothing for binarized MNIST when PCD or PA is used in the negative phase.

| Struct. | K | PCD | PA |
|---|---|---|---|
| 1 — | 1 | **89.25±0.04** | 89.35±0.06 |
| | 5 | **88.18±0.08** | **88.25±0.03** |
| | 25 | **87.66±0.09** | **87.67±0.07** |
| 1 ∼ | 1 | **84.95±0.05** | **84.93±0.02** |
| | 5 | **84.25±0.04** | **84.21±0.02** |
| | 25 | **83.91±0.05** | **83.93±0.06** |
| 2 ∼ | 1 | 83.48±0.04 | **83.37±0.02** |
| | 5 | 83.12±0.04 | **82.99±0.04** |
| | 25 | 83.06±0.03 | **82.85±0.03** |
| 4 ∼ | 1 | 83.62±0.06 | **83.18±0.05** |
| | 5 | 83.34±0.06 | **82.95±0.07** |
| | 25 | 83.18±0.05 | **82.82±0.02** |

# B    On the Gradient Variance of the Power-function Smoothing

Our experiments show that power-function smoothing performs best because it provides a better approximation of the binary random variables. We demonstrate this qualitatively in Fig. 1 and quantitatively in Fig. 2(c) of the paper. This is also visualized in Fig. 3. Here, we generate $10^6$ samples from $q(\zeta) = (1-q)r(\zeta|z=0) + qr(\zeta|z=1)$ for $q = 0.5$ using both the exponential and power smoothings with different values of $\beta$ ($\beta \in \{8, 9, 10, \ldots, 15\}$ for exponential, and $\beta \in \{10, 20, 30, \ldots, 80\}$ for power smoothing). The value of $\beta$ is increasing from left to right on each curve. The mean of $|\zeta_i - z_i|$ (for $z_i = \mathbb{1}_{[\zeta_i > 0.5]}$) vs. the variance of $\partial \zeta_i / \partial q$ is visualized in this figure. For a given gradient variance, power function smoothing provides a closer approximation to the binary variables.

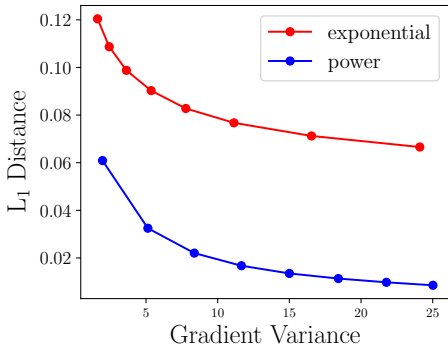

Figure 3: Average distance between $\zeta$ and its binarized $z$ vs. variance of $\partial \zeta / \partial q$ measured on $10^6$ samples from $q(\zeta)$. For a given gradient variance, power function smoothing provides a closer approximation to the binary variables.

