[Reviews · NeurIPS 2018]

Reviewer 1



Summary: -------- The authors consider variational auto-encoders with discrete latent variables under a Boltzmann machine prior. They propose two approximations to the ELBO as well as extensions to previously proposed relaxations of the discrete latents. Comments: --------- This is a solid paper that advances the methodology of training latent variable models with complex latent distributions. The paper is mostly well written (except for the comments below). It seems technically correct, and the experiments convincingly show that the proposed methods outperform previous approaches. Detailed comments: 1) Eqn 2: Why is the last term $\log p(z)$ averaged over $q(\zeta\vert x)q(z\vert x,\zeta)$ instead of just $q(z\vert x)$? In the DVAE++ it made sense as they separated into $z$ and $\zeta$ into two groups based on the RBM. 2) I found the oder of the sub-sections of section 3 confusing, which lead to some initial confusion which objective are actually used (in combination with which relaxation). Altough this was clarified later, this section would benefit from better structuring. 3) Section 3.4 is a neat applicaton of the inverse function thm. 4) Do the authors have an intuition for why the power-function approximation work better? If so, it would be valuable to include it. 5) In general, I remain undecided if discrete latent varibles are a good idea. As the authors argue, they might allow for more effiencient models and might be more interpretable; in practice however, we end up with more expensive algorithms and weaker models. So, it would be important to convince readers like myself, that discrete models have something to offer in terms of interpretibility. Have the authors investigated the learned latent representation?

Reviewer 2



# Response to author feedback Having read the authors' response I am happy with my original assessment that this would be a good addition to the conference and vote for including this submission. The authors' discussion of the potential reasons for the relatively poor performance of the Gaussian integral trick method in their feedback in response to a comment in my review seem reasonable and from my reading they also adequately address the points raised by the other reviewers. --- # Summary and relation to previous work This submission builds upon several previous pieces of work proposing methods for including discrete latent variables in variational autoencoder (VAE) models (Kingma and Welling, 2014). A key aspect of these works is retaining the ability to train the models with low-variance reparameterisation trick based gradient estimates of the variational objective by relaxing the discrete latent variables with associated continuous valued variables. Of particular significance to this submission are the discrete VAE (dVAE) (Rolfe, 2016) and dVAE++ (Vahdat et al., 2018) models which use a Boltzmann machine (BM) prior on the discrete latent variables and construct a differentiable proxy variational objective by introducing continuous variables zeta corresponding to relaxations of the discrete variables $z$, with $\zeta$ depending on $z$ via a *smoothing* conditional distribution $r(\zeta | z)$. The generative process in the decoder model is specified such that generated outputs $x$ are conditionally independent of the discrete variables $z$ given the continuous variables $\zeta$. An issue identified with the (differentiable proxy) variational objective used in both the dVAE and dVAE++ approaches is that it is not amenable to being formulated as an importance-weighted bound, with importance-weighted objectives for continuous VAE models having been found to give significant improvements in training performance (Burda et al., 2015). In this submission the authors suggest an alternative dVAE formulation they term dVAE# which is able to use an importance weighted objective. This requires evaluating (/estimating) the log marginal density on the $\zeta$ variables. In general evaluation of the marginal density on $\zeta$ is intractable as it requires marginalising $z$ from joint density on $z$ and $\zeta$, with the exhaustive summation over all $z$ configurations in general intractable to compute. The authors main contribution is to propose two alternative approaches to overcoming this issue. In the first the smoothing conditional density $r(\zeta | z)$ is assumed to take the same form as the 'overlapping smoothing transformations' previously proposed in the dVAE++ paper. In this case as the marginalisation over $z$ is intractable, the authors propose fitting a mean field approximation to the BM prior on $z$ and using this factorial approximation to approximate the marginalisation over $z$ to compute approximations to the log marginal density on $\zeta$ and its gradient. The second suggestion is to use a particularly tractable form of continuous relaxation proposed previously for sampling from BM models (Zhang et al., 2012), termed the 'Gaussian integral trick', which allows exact marginalisation over $z$, at the expense over a more complex geometry resulting marginal distribution on the $\zeta$ variables (and so difficulties in choosing an appropriate variational approximation). The authors also propose generalisations of the 'overlapping smoothing transformations' previously employed in the dVAE++ paper exploit an idea for applying the reparameterization trick to mixture distributions (Graves, 2016) to allow more general forms for the transformations while still remaining reparameterizable, with in particular several transformations which it is suggested will give lower variance gradient estimates being proposed. The proposed dVAE# approach is empirically validated to generally give significant improvements over the previous dVAE and dVAE++ approaches in terms of test set log likelihoods on two benchmark binarized image generative model datasets (MNIST and OMNIGLOT) across a range of different model architectures for each method. The benefit of using an importance weighted objective is validated by a generally increased performance as more importance samples are used and the mean-field approach using a particular 'power-function transformation' instance of their generalised smoothing transformation found to generally give better performance than the other suggested alternatives. # Evaluation This submission mainly proposes fairly incremental extensions to the existing dVAE and dVAE++ approaches, however the suggested improvements are motivated well and explained clearly, and the extensive experiments appear to show consistent significant test-set log-likelihood gains for a particular instance of the proposed approach (power-function transform with mean-field approximation) compared to competing dVAE and dVAE++ approaches on a wide range of architectures and across two different datasets. Further the inclusion of the negative results for their being significant gains to using what would seem a perhaps natural alternative of a Gaussian integral trick relaxation, given the ability to sidestep the need to introduce a mean-field approximation in this case, is also an important and interesting empirical contribution. It would be helpful however to have more discussion of potential reasons why the Gaussian integral approach seems to generally not perform as well. Overall though the originality of the proposal is not massive, the quality of the communication of the ideas present and relatively strong empirical validation of the proposals mean I feel the submission would be a worthwhile inclusion in the conference. # Minor comments / typos / suggestions: Equation 5: How to get from the second to third line is not necessarily immediately obvious - might be worth noting this relies (I think) on the linearity of the gradient of the augmented energy function with respect to z. L154: 'covariance matrix becomes diagonally dominant' -> I think this should be precision not covariance matrix? Digit grouping: in a few places commas are used for thousands digit grouping - generally a thin space should be preferred as a comma can be ambiguous as to whether a decimal point is intended in some locales. Figures 1 and 2: plots difficult to read at print size due to light weight of text / lines - it would help to plot with fonts, line weights etc. specifically set for small figure size rather than what looks like downscaling a larger plot. References: several references e.g. [3], [4]. [8], [9], [21] list arXiv pre-print versions when published versions are available. More of a nitpick, but it would also be nicer if all references consistently used initials or full first names. Hamiltonian in reference [41] should be capitalised.

Reviewer 3



The work is a further improvement of the DVAE. Most notably, the work exploits various elaborate techniques to allow the DVAE to be trained by the IWAE objective function. Specifically, on the prior distribution side, the works proposes (1) a mean-field approximation as well as (2) an alternative parameterization that enables the Gaussian integral trick. To accommodate the structure of the prior, a more general and inverse CDF free approach to reparameterizing overlapping transformations is proposed, with three different instantiations. Empirically, it is verified that with multiple samples DVAE# trained by the IWAE usually outperforms DVAE and DVAE++. Also, the power-function transformation shows a superior performance in terms of both better likelihood score and lower gradient variance. Strength: - The idea of training DVAE with the IWAE objective is well motivated. - The techniques proposed to achieve the goal are very delicate. - The empirical results support the proposed ideas. Weakness: - While the novelty is clear there, many proposed techniques are too specific in the sense that they can only be applied to the particular DVAE design with a Boltzmann prior and one-to-one correspondence between the discrete (binary) latent variable and continuous latent variable. - Due to the delicacy of the model design and training procedure, it will be quite difficult for others to replicate the work. Overall, I find the work solid, though not general enough.